# The Effect of Molecular Weight on the Solubility Properties of Biocompatible Poly(ethylene succinate) Polyester

**DOI:** 10.3390/polym13162725

**Published:** 2021-08-15

**Authors:** Mohamed M. Abdelghafour, Ágoston Orbán, Ágota Deák, Łukasz Lamch, Éva Frank, Roland Nagy, Adél Ádám, Pál Sipos, Eszter Farkas, Ferenc Bari, László Janovák

**Affiliations:** 1Department of Physical Chemistry and Materials Science, University of Szeged, Rerrich Béla tér 1, H-6720 Szeged, Hungary; m.abdelghafour2015@yahoo.com (M.M.A.); agoston.orban.99@gmail.com (Á.O.); dagota13@yahoo.com (Á.D.); 2Department of Chemistry, Faculty of Science, Zagazig University, Zagazig 44519, Egypt; 3Department of Organic and Pharmaceutical Technology, Faculty of Chemistry, Wrocław University of Science and Technology, Wybrzeże Wyspiańskiego 27, 50-370 Wrocław, Poland; lukasz.lamch@pwr.edu.pl; 4Department of Organic Chemistry, University of Szeged, Dóm tér 8, H-6720 Szeged, Hungary; frank@chem.u-szeged.hu; 5Department of MOL Department of Hydrocarbon and Coal Processing, Faculty of Engineering, University of Pannonia, Egyetem Str. 10, H-8200 Veszprém, Hungary; nroland@almos.uni-pannon.hu; 6Department of Inorganic and Analytical Chemistry, University of Szeged, Dóm tér 7, H-6720 Szeged, Hungary; adelada@chem.u-szeged.hu (A.Á.); sipos@chem.u-szeged.hu (P.S.); 7HCEMM-USZ Cerebral Blood Flow and Metabolism Research Group, University of Szeged, Dugonics Square 13, H-6720 Szeged, Hungary; eszter.farkas.szeged@gmail.com; 8Department of Cell Biology and Molecular Medicine, Faculty of Science and Informatics & Faculty of Medicine, University of Szeged, Somogyi Str. 4, H-6720 Szeged, Hungary; 9Department of Medical Physics and Informatics, Faculty of Medicine & Faculty of Science and Informatics, University of Szeged, Korányi Fasor 9, H-6720 Szeged, Hungary; bari.ferenc@med.u-szeged.hu

**Keywords:** biocompatible polyester, poly(ethylene succinate), different molecular weight, solubility study, direct condensation polymerization

## Abstract

Poly(ethylene succinate) (PES) is one of the most promising biodegradable and biocompatible polyesters and is widely used in different biomedical applications. However, little information is available on its solubility and precipitation properties, despite that these solution behavior properties affect its applicability. In order to systematically study these effects, biodegradable and biocompatible poly(ethylene succinate) (PES) was synthesized using ethylene glycol and succinic acid monomers with an equimolar ratio. Despite the optimized reaction temperature (T = 185 °C) of the direct condensation polymerization, relatively low molecular mass values were achieved without using a catalyst, and the *M_n_* was adjustable with the reaction time (40–100 min) in the range of ~850 and ~1300 Da. The obtained crude products were purified by precipitation from THF (“good” solvent) with excess of methanol (“bad” solvent). The solvents for PES oligomers purification were chosen according to the calculated values of solubility parameters by different approaches (Fedors, Hoy and Hoftyzer-van Krevelen). The theta-solvent composition of the PES solution was 0.3 v/v% water and 0.7 v/v% DMSO in this binary mixture. These measurements were also allowed to determine important parameters such as the coefficients A (=0.67) and B (=3.69 × 10^4^) from the Schulz equation, or the *K_η_* (=8.22 × 10^−2^) and α (=0.52) constants from the Kuhn–Mark–Houwink equation. Hopefully, the prepared PES with different molecular weights is a promising candidate for biomedical applications and the reported data and constants are useful for other researchers who work with this promising polyester.

## 1. Introduction

Recently, aliphatic polyesters have attracted considerable interest for research related to biomedical applications such as drug delivery systems (DDS) [1], functional materials [2], and artificial implants in tissue engineering [3] due to their preferred biodegradability and biocompatibility features, and because they are one of the most important groups of biodegradable synthetic polymers [4,5]. The main advantage of these polyesters is their biocompatibility and high ability to hydrolyze in the human body [6]. Biocompatible and biodegradable polymers have a lot of advantages in the nanotechnology field, particularly for preparing nanoparticles that can be used as DDS to protect the drug (active substance) against in vitro and in vivo degradation, which enhances the therapeutic efficacy, adjusts the releasing process, prolongs the drug activity, and reduces the side effects and the frequency of the administration of the drug [7,8].

Poly(ethylene succinate) (PES) is one of the most promising biodegradable polyesters [9] with excellent biocompatibility, especially good biodegradability. The biocompatibility of PES was comparable to the high biocompatibility polymers such as polylactic acid (PLA) and polycaprolactone (PCL), so PES is also suitable for biomedical applications as a drug carrier, similar to PCL or PLA that are already widely used [4]. It was reported that the PES shows comparable mechanical properties to polypropylene and polyethylene, in addition to numerous studies that concern the evaluation and examination of PES biodegradation under various conditions and environments [10,11,12,13,14,15,16,17]. For example, the hydrolytic degradation rate of PES is much faster than poly(butylene succinate) (PBS), making PES a more suitable substrate for drug releasing applications than PBS. Polycondensation of diols and dicarboxylic acids was early used to synthesis aliphatic polyesters that was reported by Carothers and Dorough (1930) [18]. Ajioka et al. reported that direct condensation polymerization can be used to prepare polyesters from dicarboxylic acids and diols with relative molecular weights of up to 300 kDa [19]. For the preparation of PES with a relatively high molecular weight using the two-stage melt polycondensation method (esterification and polycondensation), the PES had an intrinsic viscosity value (*η*) of 1.08 dL/g, and Tsai et al. reported a weight average molecular weight (*M_w_*) of 167 kDa [20]. However, the two-stage melt polycondensation method requires the use of a moderately toxic catalyst, such as tetrabutoxy titanium, and a relatively low molecular weight polyester formation [21,22,23], whereas direct polycondensation provides a simple and catalyst-free method for the synthesis of lower molecular weight polyester.

Moreover, the step-growth polymerization mechanism allows the synthesis of polyesters with regulated molecular mass. The average molecular weight and its distribution are important properties that determine the solubility of a polymer. The molecular weight-dependent solubility of the polymers is well-known in the literature for a long time [24]. The phenomenon of longer chains of macromolecules being less soluble than the shorter ones of the analogous structure has been known for some time and has been utilized for the separation of chains on a length basis [25]. Since Schulz suggested that fractional precipitation is useful in investigating the molecular weight and its distribution, this fractionation method was applied extensively for molecular weight determination [26]. However, this very simple and fast method requires knowledge of the corresponding constants. For the precipitation of polymer from the appropriate polymer solution, the constants of the solvent–polymer precipitant system A and B can be determined using the linearized form of the Schulz precipitability equation [26].

The solubility of polymers in various solvents have been extensively studied and numerous approaches were introduced in order to enable prediction of solvent–polymer interactions [27,28]. Most of the mentioned approaches are connected with solubility and miscibility (Flory–Huggins) parameters, measurable by both experimental methods [28] as well as calculations by group contributions approaches [27]. In general the mentioned approaches are particularly useful for different biocompatible polymers as host materials for drug delivery systems [29,30], considering both compatibility of the polymer with drug molecules as well as influence of solvent on the mentioned colloidal systems stability. Recently, the mentioned parameters were used to study PES oligomers blends with PVA polymer in both room temperature (25 °C) and melt state (200 °C) by both modified Hoftyzer–van Krevelen and molecular modeling (estimation of total solubility parameter) approaches, respectively [31].

The study aims to prepare PES with molecular mass tailored solubility properties by catalyst-free direct condensation polymerization process to use it for potential biomedical applications, which may include, e.g., biocompatible plasticizers for high molecular weight polymers toward biomedical applications [30] as well as building blocks for self-assembled nanocarriers [31]. The successful polymerization was confirmed by FTIR, while the molecular mass of the polyester obtained was determined by ^1^H NMR spectroscopy, gel permeation chromatography (GPC) and liquid chromatography–mass spectrometry (LC-MS) measurements. Because the solubility and precipitation properties of the PES were also thoroughly studied in biocompatible DMSO/water binary system, the corresponding Schulz-constants (A and B) were also calculated to recognize the polymer properties and evaluate the applicability for drug delivery system applications. The performed studies provided the crucial information about the catalyst-free single-step synthesis of poly(ethylene succinate) oligomers with narrow molecular weight distribution, controlled simply by the reaction time.

## 2. Materials and Methods

### 2.1. Materials

Succinic acid (C_4_H_6_O_4_, SA, 99%), ethylene glycol (C_2_H_6_O_2_, 99.99%) were purchased from Molar Chemicals Kft. (Halásztelek, Hungary). Sodium hydroxide (NaOH) and dimethyl sulfoxide (DMSO) were obtained from Merck (Gernsheim, Germany). Toluene and 1,4-dioxane were purchased from Molar Chemicals Kft. (Halásztelek, Hungary) and Sigma-Aldrich (Budapest, Hungary), respectively. All chemicals were used as received without further purification. Acetonitrile (C_2_H_3_N, ACN, LC-MS grade of Sigma Aldrich), Sodium hydrogen carbonate (NaHCO_3_, 99+% HPLC grade of Acros Organics (Geel, Belgium)) and purified water obtained from a Millipore-MilliQ system (Sigma-Aldrich, Budapest, Hungary).

### 2.2. Synthesis of Poly(ethylene succinate) with Different Molecular Weights

Poly(ethylene succinate) PES was prepared using the direct melt polycondensation method (Figure 1). The equimolar ratio of monomers was ensured as the following; 2.1 g of succinic acid (17.78 mmol) and 1 mL of ethylene glycol (17.78 mmol) were heated with increasing reaction times (40, 50, 60, 70, 80, and 100 min) in the oil bath at ~185 °C under continuous magnetic stirring. At selected time intervals, the quenching of polymerization reaction was performed via rapidly cooling of the reaction mixture in the ice bath. The obtained crude product was dissolved in minimal amount of THF (ca 3–4 mL per one gram), followed by precipitation with ca 15-fold excess of methanol and centrifugation. The purified polymers were dried at 40 °C for 24 h. The solvents for PES purification were carefully studied by different solubility parameter calculations approaches (Fedors, Hoy and Hoftyzer-van Krevelen)—see detailed information in the ESI file.

### 2.3. Methods of Characterization

The Fourier transform infrared spectroscopy (FTIR) measurements were performed on the succinic acid, ethylene glycol, and PES to confirm the polymerization process via using a BioRad FTS-60A FT-IR spectrometer. The spectra were recorded by the accumulation of 128 scans between 4000 and 600 cm^−1^ at a resolution of 2 cm^−1^.

Determination of the degree of polymerization was performed by determining the unreacted COOH group of succinic acid by using the acid-base titration method, 100 mg of the unpurified polymers were dispersed well in 10 mL of distilled water then the obtained solution was titrated with 0.1 M NaOH in the presence of phenolphthalein indicator, the same procedure was repeated with using the initial reaction mixture (0 min of reaction time) as reference. The polymerization conversion was calculated by using Equation (1) and the degree of polymerization (${\bar{X}}_{n})$ was calculated by monomer conversion (P) using the Carothers equation (Equation (2)).
(1)$$\mathrm{Percent}\;\mathrm{of}\;\mathrm{polymerization}\;\left(\%\right)=\frac{\mathrm{Initial}\;\mathrm{COOH}-\;\mathrm{Unreacted}\;\mathrm{COOH}}{\mathrm{Initial}\;\mathrm{COOH}}\times 100$$
(2)$$\mathrm{Degree}\;\mathrm{of}\;\mathrm{polymerization}\;\left({\bar{X}}_{n\;}\right)=\frac{1}{1-P}$$

The synthesized PESs were characterized by ^1^H NMR spectroscopy. NMR spectra were recorded with a Bruker DRX 500 instrument ((Bruker, Billerica, MA, USA)) at room temperature in DMSO-*d_6_* using tetramethylsilane (TMS) as an internal standard. Chemical shifts are reported in ppm (*δ* scale). About 1–5% w/v polymer solutions were used for the measurements. The number of repeating units (n) and the polymer molecular weights (*M*_n_) were determined by the method reported by Izunobi and Higginbotham [32].

The average molecular weight and its distribution were measured by gel permeation chromatography (Waters 2414GPC system fitted with RI detector, Ultrastyragel 1000 Å, 500 Å, 100 Å). Polyester samples were dissolved in tetrahydrofuran, then filtered with 0.2 μm Millipore (Millex filter) membrane. 1.0 mL/min eluent flow rate was used and 100 μL of 0.1% sample solution was injected. The calibration was carried out by using polystyrene standards, and a 3rd order polynomial calibration curve was described.

For the MS measurements were performed on an Agilent G6125B LC-MSD + ESI coupled with an Agilent 1260 Infinity II HPLC which was used to deliver the samples to the MSD. The samples were dissolved in LC-MS grade acetonitrile to obtain solutions with around 50 ppm concentration. The applied eluents were A: 10 mM NaHCO_3_ solution made up in purified water (15%) and B: acetonitrile (85%). The addition of NaHCO_3_ to the polar solvent was necessary to ensure the deprotonation of the carboxylate groups of the oligomers, thus enabling their MS detection in negative mode. The measurements were carried out in negative mode, in scan mode monitoring the 300–1750 m/z range. The average molecular weight was then calculated from the relative abundances and the individual masses.

Based on the change in the molecular weight of the prepared polyesters, the solubility or precipitation properties of synthesized polyesters in the DMSO/H_2_O binary system were studied as the following; 100 mg of polyesters (with different molecular weights) was dissolved in 10 mL of DMSO as a good solvent. During the measurements, the precipitation of the obtained polyester solutions as a function of the dropwise addition of water (as poor solvent) was monitored with an ISO Portable Turbidity Meter-HI98703 (Hanna instruments) under continuous magnetic stirring at 25 °C.

The fractional precipitation method was also used to determine the average molecular weights of PES and the constants of the Schulz-equation (Equation (3)) characteristic of the solvent–polymer-precipitant system A and B [26]. For this purpose, PES 80 min was chosen. The essence of the method is that PES was dissolved in DMSO and precipitated with distilled water, then the corresponding molecular weights of the separately collected fractions can be measured with an independent method (e.g., NMR or DLS). During the fractional precipitation, the distilled water (as precipitant) was added dropwise to 100 mL of 5% w/v of polyester solution (in DMSO) using a burette, and different fractions were collected. The precipitated polyester was centrifuged at 15,000 rpm for 30 min at 25 °C. The supernatant was discarded and the differentiated polymer fractions were dried in an oven followed by measurement of the relevant molecular weights.
(3)$$\frac{1}{M}\;=\frac{100}{B}\;\Phi -\;\frac{A}{B}$$

Plotting the reciprocal of the molecular weight (1/M) of a given sample as a function of Φ (volume fraction of distilled water as poor solvent or precipitant), and the slope of this line gives the value of B and then A can be calculated from the value of B and the knowledge of the intercept value. After the determination of these coefficients, the fractional precipitation method was also used to determine the average molecular weights of PES based on the Schulz equation and knowing the values of the A and B constants.

Differential scanning calorimetry (DSC) measurements were used to investigate the thermal behavior of the prepared polyesters using a Mettler-Toledo DSC822e instrument. Initial monomers and the obtained polyesters were used for the measurements; the samples were heated from 25 to 500 °C with a heating rate of 5 °C/min. Thermogravimetric (TG) analysis was used to study the heat degradation and thermal behavior of PES-80 min using a Mettler-Toledo TGA/SDTA 851e instrument. The heating rate of the sample was 5 °C/min from 25 to 500 °C.

The theta (Θ) compositions of mixed solvents for polyester were estimated from the turbidimetric titration that was introduced by Elias and then modified by Cornet and van Ballegooijen [33]. Water as non-solvent was added dropwise to PES-80 min solution in DMSO under continuously magnetic stirring until the beginning of the phase separation (precipitation) at room temperature and the turbidity was monitored with an ISO Portable Turbidity Meter-HI98703 (Hanna instruments). This procedure was performed with different concentrations (0.2–1 w/v) of the polyester (PES-80 min) solution. By plotting the logarithm of volume fraction of water (non-solvent) needed to incipient precipitation as a function of the logarithm of the corresponding volume fraction of polyester, the Θ-composition can be determined from extrapolation to pure polyester (Φ_PES_ = 1) [33].

Based on the Kuhn–Mark–Houwink-(KMH) equation (Equation (4)), the coefficient (*K_η_* and exponent α were determined using a series of PES samples with different molecular weights and measured the relevant intrinsic viscosity (*η*) values. The intrinsic viscosity for each PES sample was determined by measuring the viscosity (proportional to the flow times, Equation (5)) of PES solution (in DMSO) with the variation of the concentration that was ranged between 0.1–0.3 g/mL using the Ostwald viscometer at room temperature, then the reduced viscosity (*η*_red_) that was determined by using (Equation (6)) was plotted against the concentration (g/mL) and the intrinsic viscosity (*η*) was estimated from the intercept value. The logarithm of determining intrinsic viscosity (*η*) for three different PES was plotted against the logarithm of relative molecular weights, and the *K_η_* was obtained from the intercept value and exponent α was determined from the slope from the curve fitting equation [34].
(4)$$\left[\eta \right]=KM^{\alpha }$$
(5)$$\eta_{sp}=\frac{t-t_{o}}{t_{o}}$$
(6)$$\eta_{red}=\frac{\eta_{sp}}{c}$$
where *t* is the flow time of the solution, *t_o_* is the flow time of pure DMSO, *c* is the concentration, *K_η_* is the KMH coefficient relevant to a given polymer-solvent system, and α is the KMH exponent which characterizes the conformation of macromolecules in a solvent [34].

## 3. Results and Discussion

In an attempt to produce a suitable biocompatible and biodegradable polyester that can be potentially used for drug delivery applications, PES was synthesized using a catalyst-free direct condensation polymerization method with polymerization time depending molecular weight. Thus, the direct condensation polymerization between succinic acid (dicarboxylic acid) and ethylene glycol (diol) in an equimolar ratio at 185 °C (Figure 1) was performed without using catalysts (green chemistry) since the main goal of this work is synthesizing polyester to be suitable for use in biomedical applications such as drug delivery or tissue engineering. The appearance of a yellowish, waxy product (Figure 1 inserted photo) from the transparent reactants was indicated the polycondensation of the initial monomers and the formation of the polyester macromolecules. Moreover, because of the planned biomedical applications, we also paid attention to the purification of the synthetized PES samples since the purification method based on simple dissolution/ precipitation of polymer (see calculation of solubility parameters for PES oligomers and organic solvents in ESI and the results in Appendix A) enabled to obtain high purity products. In our studies solubility parameters were used to choose appropriate solvents for PES polymers purification. In general, the good solvent should be thermodynamically compatible with the polymer, in contrast to the bad solvent. Although it is possible to choose the solvents experimentally, but the mentioned approach may not fulfill the requirements of the purification process, when the polymer is prone to form thermodynamically instable solutions in both good and bad solvents. According to the calculated data presented in the ESI file, THF and methanol were found to be appropriate thermodynamic “good” and “bad” solvents, respectively, for PES oligomers, obtained for different polycondensation times. Both solvents are freely miscible with water as well as are characterized by low boiling points (below 70 °C).

In our work, controlling of molecular weight of the prepared PES was carried out by changing the polycondensation times (40, 50, 60, 70, 80, and 100 min) of this step-growth polymerization reaction mechanism [35,36]. The successful polycondensation was confirmed by FTIR measurement. Figure 1 shows the FTIR spectra of succinic acid, ethylene glycol, and synthetized-PES (after 80 min polycondensation time).

The main characteristic peaks of succinic acid are 3200–2500 cm^−1^ correspond to the broad stretching vibration of the O–H group, 2640 and 2540 cm^−1^ are due to C–H stretching, while the peak at 1690 cm^−1^ corresponds to C=O group stretching vibration and peak at 900 cm^−1^ results from the out of plane bending of the bonded -OH group of the carboxylic acid. The peaks at 1420 and 1305 cm^−1^ were due to C–O–H in-plane bending (δ_C-O-H_) and C–O stretching vibration, respectively [37]. The main characteristic peaks of ethylene glycol are 3380 cm^−1^ due to the OH group stretching vibration and the peaks at 2940 and 2880 cm^−1^ were assigned to asymmetric and symmetric of C–H stretching. The peaks at 1460 and 1410 cm^−1^ result from CH_2_ bending and C-O-H bending, respectively. The peaks at 1100 and 1050 cm^−1^ are due to C–O stretching, while the peak at 880 cm^−1^ for CH_2_ rocking [38]. FTIR spectrum of PES shows that the appearance of a peak at 1720 cm^−1^ that characteristic of the C=O group stretching vibration of the ester bond and disappearance of C=O group stretching vibration of succinic acid and OH group stretching vibration of ethylene glycol that confirms the success of the polymerization reaction. Peaks in the range of 1150–1220 cm^–1^ were attributed to the stretching vibration of the (–C–O–C–) group in the ester bond of PES. The peaks at 1040 and 920 cm^−1^ were assigned to (–O–C–C–) stretching vibrations and (–C–OH) bending in the carboxylic acid groups of PES, respectively. Additionally, the peaks at 2960 and 1388 cm^–1^ correspond to asymmetric stretching vibration and symmetric deformational vibrations of –CH_2_– groups in the main chain of PES, respectively [39].

Thermal properties of the synthesized PES samples and initial monomers were also studied by using DSC and TGA measurements. Figure 2A shows the DSC curve of the used initial monomers, while Figure 2B indicates the DSC (grey) and TG (black) curves of PES (80 min) to prepare the PES as a reference to verify the absence of monomers in the prepared PES samples. From Figure 2A it can be seen that the melting point of succinic acid is 184 °C, while the boiling point of ethylene glycol and succinic acid monomers are at 205 and 255 °C, respectively. Based on the decomposition temperature and boiling point of monomers used, the ideal polymerization temperature should be between 184 and 195 °C to get melted succinic acid and to avoid the decomposition of monomers and the obtained polyesters. Figure 2B shows the comparison of the TG curve and DSC curve for PES 80 min, the melting point (T_m_) of PES 80 min was ~80 °C and the crystallization temperature (T_c_) of the PES was significantly observed at around 55.3 °C [40], the decomposition of polyester started after ~210 °C in both curves indicating that the first decomposition step was between 215 to 315 °C with ~25% weight loss, and the second decomposition step was between 315 to 500 °C with a weight loss of approximately 65% and total weight loss was ~90%. Based on these results, it is not recommended to use a polymerization temperature higher than 200 °C during the polyester synthesis to avoid decomposition of polyester and decreasing the molecular weight by the thermal degradation process.

After the determination of the ideal polycondensation temperature (185 °C), the polymerization yield of the prepared PES was measured at this temperature by using simple acid-base titration to determine the reacted amount of succinic acid and compared with the initial amount and presented as a function of polymerization time as shown in Figure 3. The percent of polymerization was gradually increased with increasing the reaction time due to the step-growth mechanism through the formation of dimes, then trimers, oligomers, and eventually long-chain polymers. Increasing the reaction time led to an increase in the chance of reaction between OH groups of ethylene glycol and COOH groups of succinic acid, which means gradually decreasing the available COOH groups over time until reached to maximum polymerization percentage at around 92% after 80–100 min of reaction time.

Next, we also studied how the molecular mass changes with the increasing polycondensation time. The chemical structure and the molecular weight of PES was characterized by ^1^H NMR spectroscopy. Since the polymer was synthesized by the reaction of the equimolar amount of monomers (i.e., succinic acid and ethylene glycol), it is expected that the resulting PES will terminate (statistically) at one end with a carboxylic group (end group I) and the other end with a hydroxyl group (end group II). The number of the repeating units (*n*) and thus the number-average molecular weight (*M_n_*) of the PESs were calculated by the integrated intensities of the characteristic proton peaks [32]. Figure 4 shows the partial ^1^H NMR spectrum of PES-80 min with peak assignments as well as the chemical structure and the calculated molecular mass of the end groups and the repeating unit. The singlet peaks appearing at 4.21 (*δ*H*^c^*) and 2.57 (*δ*H*^d^*) ppm were assigned to the methylene protons *c* and *d* in the repeating units of PES [41]. The triplets of CH_2_ protons of end group II can be observable at 4.02 (*δ*H*^e^*) and 3.55 (*δ*H*^f^*) ppm, respectively, with lower intensity, while the methylene protons of end group I give overlapping signals (*δ*H*^a^* and *δ*H*^b^*) with that of the applied solvent between 2.46–2.53 ppm. The *M_n_* of PESs can be determined by the following equation (Equation (7)):(7)$$M_{n}=144\times \;\frac{I\left(c\right)}{2\times \;I\left(f\right)}+101+61$$
where *I*(c) and *I*(*f*) are the peak areas of the corresponding methylene protons, the rounded values 144, 101 and 61 are the molecular weights of the repeating unit and the end groups I and II of PES, respectively (Figure 4).

The equation also takes into account that the number of *c* protons in the repeating unit is twice as many as the number of *f* protons in end group II. This also means that the number of repeating units in PES-80 min is n = 8 (Table 1). The data in Table 1 are indicated that relatively low molecular weight polyesters (from *M_n_* = 846 to *M_n_* = 1312 g·mol^−1^) were synthetized without the using of catalyst, but we can also see that the *M_n_* values were increased with the increasing polycondensation time until 80 min polymerization time. Further increasing the reaction time, however, the measured *M_n_* was slightly decreased (1110 g·mol^−1^), indicating that the optimal polycondensation time is about 80 min and beyond this reaction time undesired thermal degradation was occurred even so at optimized (T = 185 °C, see Figure 2) temperature. Beside the above presented *M_n_* values obtained by NMR, the number (*M_n_*) and weight average molecular weights (*M_w_*) of the polyesters were also determined by GPC. The data are summarized in Table 1, whereas Appendix A shows a representative GPC molecular weight distribution curve obtained for polyester with 80 min polycondensation time. As it can be seen the *M_n_* values determined by GPC are higher (~2592–3394 g·mol^−1^) than the data obtained from ^1^H NMR end-group analysis (846–1312 g·mol^−1^). This difference on *M_n_* values of the same polymers obtained by GPC and NMR measurements reported several times in the literature [42,43,44]. Although GPC is the most widely used method for the determination of molecular weight distribution of polymers, the strong dependence of its data on the calibrant (here polystyrene) remains an encumbrance. For this comparison to be reasonable, it may be necessary to assume that the hydrodynamic radii of the measured polymer and the standard are the same at equivalent molecular weight. Furthermore, it may be also be necessary to assume that the polymers have similar elution volume/chain length characteristics while in the linear selective permeation regime of the GPC calibration curve. In contrast, NMR is an absolute method yielding reliable information regarding polymer microstructure and molecular weight. Therefore, the *M_n_* values determined by NMR were used in further evaluations but these data were also confirmed by LC-MS measurements.

Appendix A shows a representative MS-spectrum of the measured PES sample (80 min polycondensation time), while the obtained molecular weight values are summarized in Table 1. It can be observed that the values determined by LC-MS measurements (983–1025 g·mol^−1^) are in good agreement with the values obtained by NMR (846–1312 g·mol^−1^). On the other hand the *M_n_* values, obtained by LC-MS, do not exhibit time dependent relationship, in contrast to ^1^H NMR provided ones. The mentioned effect is possibly linked to narrow *M_n_* values distribution and relative stabilities of ionized specimens during MS analysis and detection (see Appendix A).

Additionally, the *M_w_* values were also increased (from 4287 to 5046 g·mol^−1^) with the increasing polycondensation time (from 40 to 80 min), while the polydispersity index values (PDI, ratio of *M_w_* to *M_n_*) were indicated that relatively monodisperse polymers were obtained.

Figure 5 presents the relationship between the degree of polymerization (${\bar{X}}_{n}$) that was estimated from Carothers equation (Equation (2)) as a function of monomer conversion percent (P%), from the shape of that curve we can confirm the present mechanism of polymerization for succinic acid and ethylene glycol is the step-growth mechanism where the degree of polymerization increase steadily with increasing of conversion percent until reached to ~75% after that the degree of polymerization dramatic increase until reached to the maximum degree of polymerization (12.67) at ~92% conversion percent.

The encapsulation of drug molecules with different solubility properties in polymeric matrix was often achieved by traditional (nano)precipitation, which requires the knowledge of the polymer (and drug) solution behavior properties. Thus, the solubility and precipitation properties of the synthesized PES were examined in detail using water (precipitant) and biocompatible DMSO as good solvent (see 3. chapter “Estimation of the volume fraction of water in DMSO:water mixtures for PES precipitation—the theoretical approach” in ESI). First, the Θ-composition of these biocompatible solvents was determined for the PES by turbidimetric titration. Water was added to different concentration solutions of PES-80 min (1, 0.8, 0.6, 0.4, and 0.2% w/v) in DMSO, the precipitation of polymer was monitored by turbidimeter to determine the volume fraction of water at which the polyester started to be precipitate. Figure 6 shows the logarithm volume fraction of water verse the logarithm of the corresponding volume fraction of polyester and the intercept was −0.5099 which means the theta solvent for PES is around 0.309 of water volume fraction in the water/DMSO mixture that was in complete agreement with the data presented in Figure 7.

After the determination of the Θ-composition of polyester solution in water/DMSO, we also want to know how the precipitation properties of the polymer depend on its molecular weight. It is well-known from the literature that the solubility properties of the polymers depend on their molecular weight, therefore, the solubility of the polyester samples with increasing molecular mass was also studied by precipitation method as shown in Figure 7.

According to Figure 7, the highest molecular weight polyester (~1312 g·mol^−1^) needs the lowest amount of water (Φ_water_ = 0.286) to precipitate that was also completely fitted with the theoretical study as shown in Appendix A (studies for polymer with large molecular weight and negligible effect of ending groups), while for PES with the lowest molecular weight, the precipitate appears when the amount of the added water exceeds around 0.44 (v/v). In other words, there is a reversible relationship between the molecular weight of PES and the volume fraction of water (Φ_water_). Based on both Table 1 and Figure 7, these data are fully compatible with the idea suggested here, namely the solubility of the PES can be varied by its molecular mass. Due to the changes in the molecular weight, there are obvious changes in the maximum turbidity values after adding 10 mL of distilled water to polyester solutions in DMSO, the turbidity values of PES 80 min (~1312 g·mol^−1^) and PES 40 min (~846 g·mol^−1^) were ~400 NTU and 130 NTU, respectively. The precipitation of the highest molecular weight macromolecules increases the polymer solution’s turbidity. This phenomenon is appropriate to monitor the precipitation of polymer and determine the molecular weight. Thus, it can be concluded that under the above reaction condition (T = 185 °C, t_polycondensation_ = 40–80 min, catalyst free equimolar reaction) the molecular weights of polyester were varied between *M_n_* = ~846 and 1312 g·mol^−1^, but even so on this relatively narrow range of molecular weights, the solubility/ precipitation properties of polyester were significantly changed in the DMSO/ water mixture between Φ_water_ = 0.28 and 0.44. In other words the measured *M_n_* values were increased with the increasing polycondensation time and it is also manifested in the polymer’s solubility. The catalyst-free conditions provides low molecular weight oligomers (*M_n_* values not exceeding 1312 Da by ^1^H NMR) most possibly due to reversible character of esterification reaction (hydrolysis may also occur, especially when terminal groups are not protected by appropriate active form of catalyst) as well as degradation of polymer for longer synthesis times (exemplified by PES-100).

Knowing the *M_n_* (Table 1) and Φ_water_ (Figure 6) values it is also possible to determine the corresponding A and B coefficients from the Schulz-equation (Equation (3)) [26]. According to the Schulz-theory, this method is also suitable for the determination of molecular weight if we know the relevant coefficients. In order to determine these characteristic A and B constants of the solvent–polymer-precipitant system, the linearized form of the Schulz equation (Equation (3)) was used. Figure 8 shows the plot of 1/M as a function of volume fraction of water (Φ_H2O_) for the different PES samples (40, 60, 70, 80 min). From the plot the constant values of B and A were calculated and the relevant data was recorded in Table 2. These data enable the determination of PES molecular mass by simply precipitation method.

To determine the KMH coefficient (*K_η_*) and exponent (α) according to the Kuhn–Mark–Houwink-equation (Equation (4)), the viscometry measurements were carried out in the PES with different molecular weights to determine the relatively intrinsic viscosity as shown in Figure 9A. According to Figure 9A, we can see the influence of the molecular weight of PES on the viscosity values and the tendency of increasing viscosity with increasing the PES concentration in DMSO.

The theory of solubility and miscibility parameters may help in explanation of polymers’ solutions rheological behavior. In general comparison of hydrogen bonding components (δ_h_) of solubility parameter for the solvent and the polymer may be helpful for finding indicate the theta solvent conditions (exponent α in Kuhn–Mark–Houwink equation value—0.5). In such solvents, the value of the *K_η_* constant is dependent on the dispersion (δ_d_) and polar (δ_p_) components of their solubility parameters. For the PES polymer and DMSO, the δ_h_ components of the solubility parameters are both equal to around 10 MPa^0.5^, so it is possible that DMSO constitutes a theta or near-theta solvent for PES [27]. On the other hand, low viscosity of PES solution in DMSO may be simply connected with the relatively low molecular weight (around 850–1300 Da) of poly(ethylene succinate) oligomers. That is why the analysis of solubility and miscibility parameters may only support the experimental approach. The viscosity of the polymer solution may be predicted according to the Kuhn–Mark–Houwink equation and the dependence of α constant on solubility parameters. The theoretical consideration (see ESI file) is in good agreement with the experimental values of *K_η_* and α constants for PES in DMSO (see Figure 9B), found to be equal to 8.22 × 10^−2^ and 0.52, respectively. The value of α that characterized the macromolecules conformation in the solvent was around 0.52 in the same range of 0.5 < α < 0.8 to a flexible chain [34]. The mentioned findings indicate that DMSO constitutes a near theta solvent for PES polymer and thus the DMSO/water binary mixture is ideal for the precipitation of the polymer which enables the drug encapsulation.

Our study shows that the biodegradable and biocompatible PES has different physical and chemical properties depending on relative molecular weight, and the theoretical and experimental results are fully consistent with each other, which shows promising findings that make it suitable to be used as the drug delivery system to prolong drug release through molecular weight adjustability. The latter property is particularly significant for self-assembling structures, such as polymeric micelles, where relatively short hydrophobic, polyester chains play a crucial role in tuning the hydrophilic–hydrophobic balance for optimal nanocarrier behavior.

## 4. Conclusions

Biodegradable and biocompatible PES polyester was synthesized by using direct condensation polymerization between succinic acid (dicarboxylic acid) and ethylene glycol (diol) monomers in an equimolar ratio at 185 °C. According to the FTIR, the successful polymerization reaction was confirmed by the appearance of a peak at 1720 cm^−1^ that was characteristic of the C=O group stretching vibration of the ester bond and disappearance of the C=O group stretching vibration of succinic acid and OH group stretching vibration of ethylene glycol, and the maximum polymerization yield was 92% for PES after 80–100 min polycondensation time. The appropriate THF solvent and methanol precipitant for PES purifications were chosen according to careful solubility parameters studies in order to fulfill the requirements of the thermodynamical solubility and insolubility of the PES oligomers. The molecular weight (*M_n_*) of the obtained PES samples determined by ^1^H NMR analysis was ranged between ~850 to 1300 Da depending on the polycondensation time, and it also affected the solubility properties of the polyester. The highest molecular weight PES shows the lowest volume fraction of water (Φ = 0.286) needs to complete the precipitation from the DMSO solution and vice versa. The thermal properties of the obtained polyesters were studied by using DSC and TG measurements. The prepared polyester had a melting point (T_m_) at around 80 °C with a complete absence of corresponding peaks of monomers that were also confirmed by FTIR measurement; the polymerization temperature should be under 200 °C to avoid the thermal degradation of PES. Eventually, the prepared polyester has promising physical and chemical properties that promote its use as a good candidate for drug delivery applications.

## Figures and Tables

**Figure 1 polymers-13-02725-f001:**
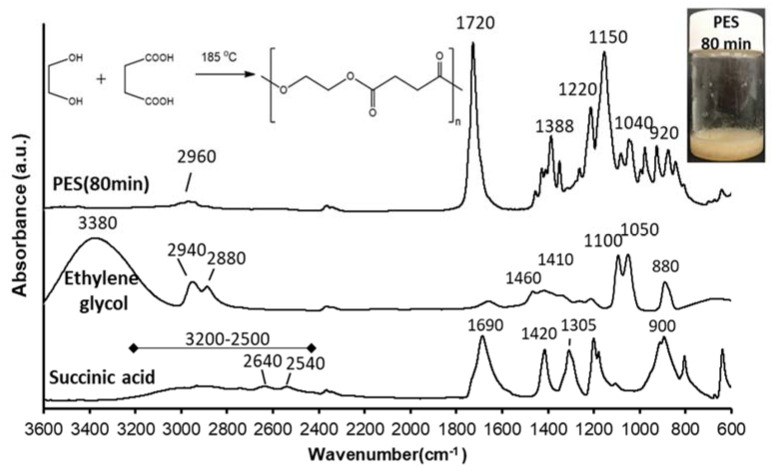
FTIR spectra of succinic acid and ethylene glycol monomers as well as the synthesized polyester sample (after 80 min polycondensation time). The inserted scheme shows the synthesis of PES by condensation reaction between ethylene glycol and succinic acid.

**Figure 2 polymers-13-02725-f002:**
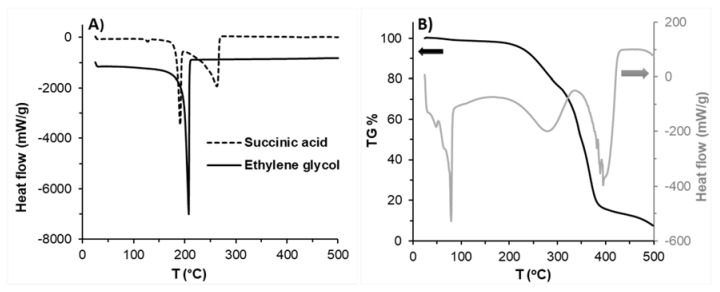
(**A**) DSC curves of the initial monomers (ethylene glycol and succinic acid), (**B**) DSC- and TG-curve of PES polyester (80 min polycondensation time).

**Figure 3 polymers-13-02725-f003:**
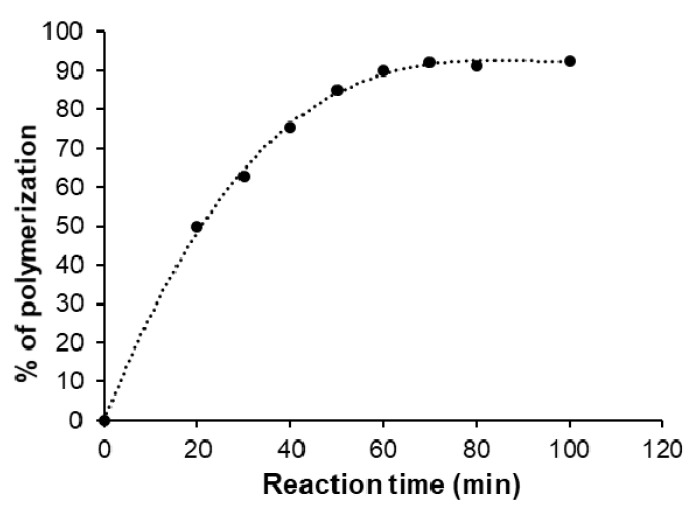
The evolution of polymerization yield as a function of polycondensation time.

**Figure 4 polymers-13-02725-f004:**
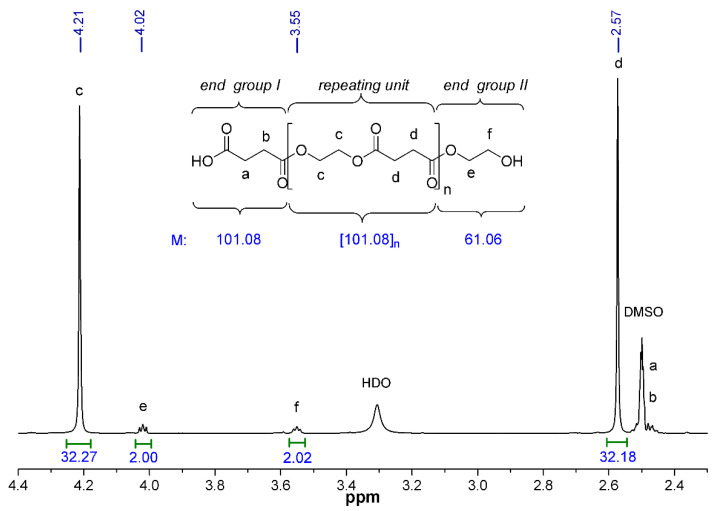
Enlarged partial ^1^H NMR spectrum (DMSO-*d*_6_, 25 °C) of PES-80 min with peak assignments and calculated molecular weights of the structural units.

**Figure 5 polymers-13-02725-f005:**
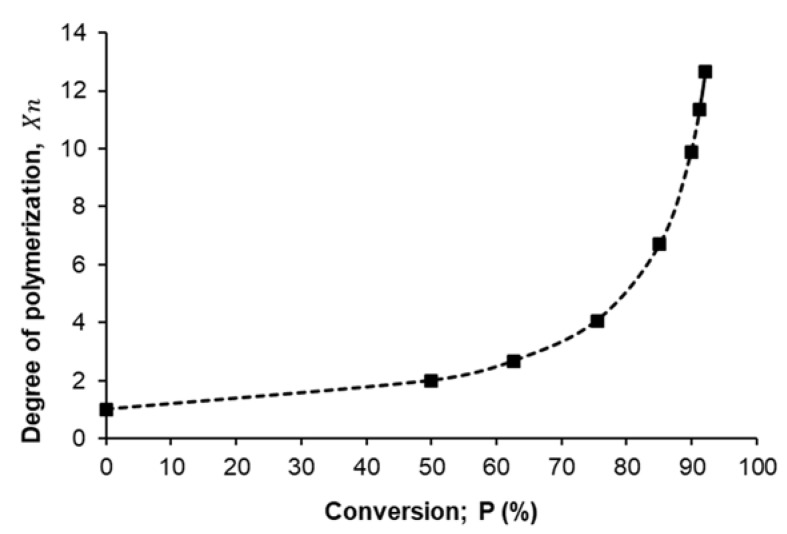
Degree of polymerization (*X_n_*) as a function of monomer conversion (P %).

**Figure 6 polymers-13-02725-f006:**
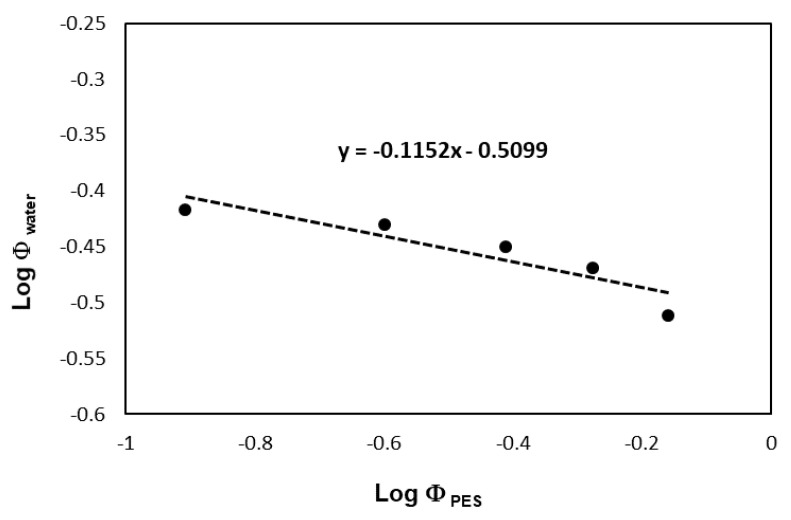
Turbidimetric titration of DMSO-based PES solution (80 min polycondensation time) with water; log volume fraction of water (Log Φ_water_) needed to incipient precipitation as a function of the log of the corresponding volume fraction of polyester (Log Φ_PES_).

**Figure 7 polymers-13-02725-f007:**
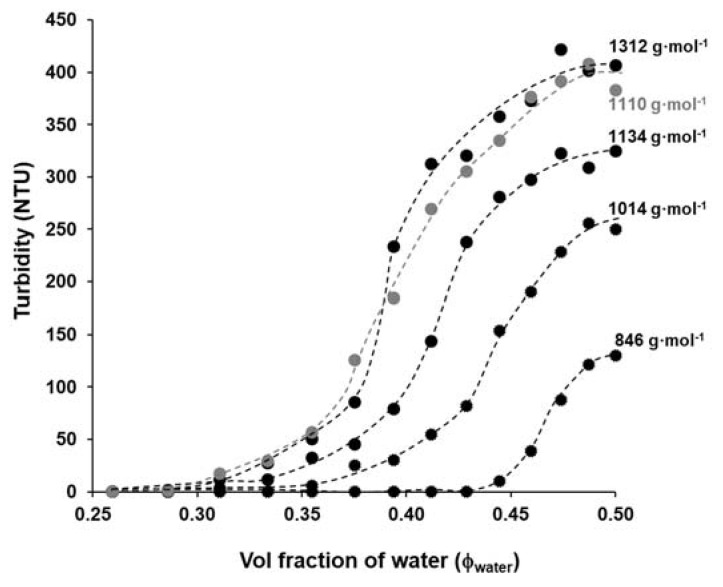
The precipitation curves of the synthesized polyester with increasing molecular weight as a function of water content in water/DMSO mixture (10 mL 1% DMSO based PES solution was precipitated by drop wise addition of distilled water).

**Figure 8 polymers-13-02725-f008:**
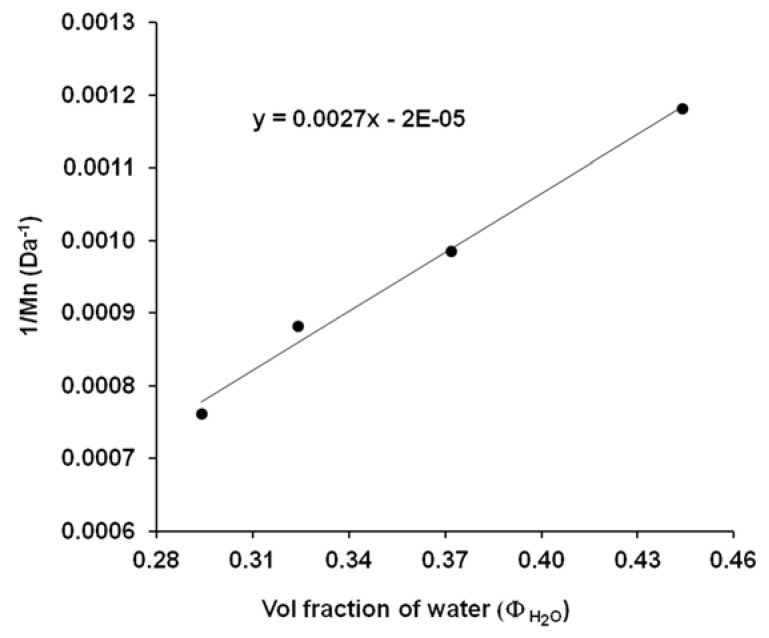
The plot of reciprocal molecular weight values (1/*M_n_*) (from NMR measurements) as a function of water volume (Φ_H2O_) in water/DMSO mixture for different PES samples (PES 40’, 60’, 70’, 80’).

**Figure 9 polymers-13-02725-f009:**
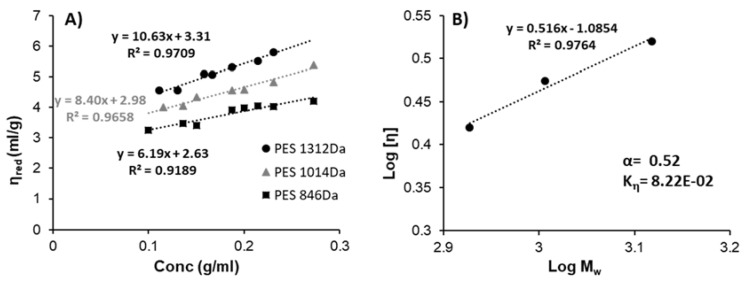
(**A**) The reduced viscosity as a function of PES with different molecular weights to determine the intrinsic viscosity, (**B**) The logarithm of intrinsic viscosity as a function of the logarithm of corresponding molecular weight to estimate the KMHS coefficient (*K_η_*) and exponent α.

**Table 1 polymers-13-02725-t001:** Degree of polymerization, *M_n_* and *M_w_* values obtained from different methods and polydispersity index for the synthesized PES samples.

Polymer	*I* (*c*) *	*I* (*f*) *	Number of Repeating Units (n)	*M_n_* * (g · mol^−1^)	*M_n_* ** (g · mol^−1^)	*M_w_* ** (g · mol^−1^)	*M_w_/M_n_* **	*M_n_* *** (g · mol^−1^)
PES-40 min	20.03	2.11	5	846	2592	4287	1.653	1009
PES-60 min	24.27	2.05	6	1014	2819	4462	1.584	1025
PES-70 min	28.21	2.09	7	1134	3158	4675	1.480	1000
PES-80 min	32.27	2.02	8	1312	3394	5046	1.513	983
PES-100 min	28.18	2.14	7	1110	2964	4466	1.507	939

* Obtained by the integrated intensities of the corresponding peaks in the related ^1^H NMR spectra; ** Determined by GPC (see ESI file); *** From LC-MS data (see ESI file).

**Table 2 polymers-13-02725-t002:** The molecular mass and volume fraction data for the calculation of A and B constants for Schultz-equation.

Polymerization Time (min)	*M_n_* (Da)	1/*M_n_*	Φ_H2O_	Data from Figure 8	Constants
40	846	0.00118	0.444	slope: 0.002709 intercept: −1.814 × 10^−5^	B = 36913 A = 0.67
60	1014	0.00099	0.372		
70	1134	0.00088	0.324		
80	1312	0.00076	0.294		

## Data Availability

Not applicable.

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
