# Peer review of "The Effect of Molecular Weight on the Solubility Properties of Biocompatible Poly(ethylene succinate) Polyester"

_polymers, 2021, doi:10.3390/polym13162725_

Round 1
Reviewer 1 Report
The manuscript submitted by Mohamed M. Abdelghafour et al reported on the “The effect of molecular weight on the solubility properties of biocompatible poly(ethylene succinate) polyester” In this work, the corresponding polymers were successfully synthesized and characterized and the solubility properties were studied theoretically and practically. But some of the key issues have to be addressed and the manuscript has to be modified before publication.
Major comments:
As a polymer related research, there was no Mw (Weight average molecular weight) and PDI (Dispersity) mentioned anywhere in the paper and no GPC/SEC data were showed. In this situation, it is impossible to rationalize the effect of molecular weight on the solubility properties when the dispersity of the polymer is unknown. In an extreme example, if the polymer with Mn= 1000 consist of 50% of 500 and 50% of 1500, it would behave very different from the polymer consist of 100% Mn=1000 in terms of solubility. Mw, PDI and GPC/SEC data are mandatory in this study, please add those data.
The authors listed and tested 5 different polymers from Mn=846 to Mn=1312 and the medium three polymers share very similar molecular weights. Frankly speaking, it is a super narrow range of molecular weights for polymers. If you want to discuss about the relationship between molecular weight and properties, you must expand this narrow range at least from Mn=846 to Mn= 5000. Otherwise it is hard to say that this topic and the title of the manuscript are well discussed and studied.
Minor comments:
The first sentence of introduction depicts a general usage of aliphatic polyesters and there are only two references attached. It seems too little as many key words were mentioned such as functional materials and artificial implants. At least cite several review papers because those concepts are not created by yourself.
The last paragraph of the introduction looks smaller than other paragraphs maybe because it used different font. Please correct that. Check the grammar and the way of writing because there are too many sentences beginning with “something were also studied/determined...” in this paragraph. Please rephrase them and make them concise and clear.
Please replace the figure 4 with a high-resolution figure. It is hard to see the NMR integrals or any words in it. It is too blurred.
Author Response
The Reviewers’ comments are always followed by our response highlighted in yellow in the uploaded file.
Reviewer 2 Report
Dear Editor, in the present work poly(ethylene succinate) (PES) was synthesized and the effect of its molecular weight on solubility and precipitation properties has been evaluated. The paper is well organized and contains some new and interesting data. For this reason, I propose to accept it for publication. In following you can find some minor importance comments.
Abstract should be without any sessions (please see the published works in Polymers and journal instructions). May I suppose that it was remaining from a previous submission to another journal?
The described applications of biobased polymers in introduction is limited and thus should be extended to many other fields. Please see a recent review and enhance this description Polymers 13, 1822, 2021. https://doi.org/10.3390/polym13111822.
Almost all prepared MW are very close (Table 1). Was any particular reason that MW not higher than 1312 g/mol were not produced?
Author Response

(The authors gave the same response as above.)

Round 2
Reviewer 1 Report
Most of the points were addressed and the manuscript can be accepted.